# Online Learning and Teaching during the COVID-19 Pandemic in Higher Education in Qatar

**Noof M. AlQashouti** [1,2], **Mohammed Yaqot** [1] and **Brenno C. Menezes** [1,*]

1   Division of Engineering Management and Decision Sciences, College of Science and Engineering, Hamad Bin Khalifa University, Qatar Foundation, Doha 34110, Qatar; noal27989@hbku.edu.qa (N.M.A.); moyaqot@hbku.edu.qa (M.Y.)
2   Ministry of Education and Higher Education, Doha 35111, Qatar
*   Correspondence: bmenezes@hbku.edu.qa

**Abstract:** The COVID-19 pandemic instigated a sudden upheaval in the realm of education, significantly impacting students and educators across the globe. The primary objective of this study was to delve into the experiences of both students (in the learning) and educators (in the teaching) amidst the emergency shift to online education within the higher education system during the COVID-19 pandemic, taking Qatar as a case study. This research adopted a quantitative approach, utilizing surveys as the primary data collection instrument. Data were collected separately from both students and educators, with a sample size of 660 students and 103 educators participating. In both surveys, a five-point Likert scale was employed to record responses. The principal findings of this research indicated that both students and educators exhibited a high level of agreement when it came to their perceptions of the digital tools employed during the educational transition and the accompanying support provided. Conversely, educators expressed a moderate level of agreement concerning their perspectives on learning materials and examination methods. The findings underscore the urgent need for higher education institutions (HEI) to strategically leverage digital tools. Recognizing the centrality of technology, this strategic utilization becomes imperative to foster a supportive and resilient learning environment for future uncertainty scenarios of lockdown and social distancing.

**Keywords:** higher education; online/hybrid education; education platform; digital tools; resilience; perception





## 1. Introduction

The education system has undergone various transformations that have improved the quality of learning and expanded the range of educational opportunities. In the past four years, the education sector has faced numerous challenges due to the COVID-19 pandemic, necessitating the adoption of online classes. In the COVID-19 outbreak, digital technology has become a primary means of education for institutions and students. In most parts of the world, mandates have created challenging situations for educational institutes, making it difficult to continue face-to-face classes and interact with students [1]. Such a crisis has revolutionized the methods of teaching and learning by embracing digitalization. It compelled educational institutions (EIs) at all levels, from primary schools to graduate research programs, to rapidly implement significant changes. The primary challenge faced by the education system was the transition from traditional classrooms to online classes. Virtually all EIs around the world, impacted by the pandemic, shifted towards online learning. Educators delivered their lessons and assessments through various online platforms, i.e., Webex, Microsoft Teams, and Zoom, among others. As technology continued to evolve, EIs adapted by devising strategies to ensure the continuity of the learning process. Distance learning presented challenges for both students who were not accustomed to the online mode and educators who needed time and preparation to adapt to the forced adoption transition from traditional in-person lectures to a virtual format [2].

This adaptation required not only technological readiness but also adjustments in behavior and procedures, and it was a learning curve for both students and educators. Al Nuaimi et al. [3] discussed the monitoring and evaluation of in-person and online outreach programs using various indicators, including participant numbers, engagement levels, satisfaction surveys, and articulation. The pandemic significantly boosted the acceptance and utilization of online education. Despite the findings by Koh and Daniel [4], which indicated that the approaches employed by educators and students to transition to online learning during the pandemic helped to maintain educational continuity and assessment practices, e.g., exam scores, several unresolved issues still persist. These issues include (1) students not experiencing an enhanced learning environment, (2) challenges in developing and implementing robust assessment methods and online examinations, (3) unequal access to digital technology, and (4) an increased workload for some faculty. The risk of social or financial inequality among students and varying levels of computer knowledge or ability among educators posed another complex issue [5]. Additionally, there were questions regarding the extent of student engagement in such courses [6]. Tudorie et al. [7] highlighted technology's potential as a pivotal tool, not just in times of crisis, but also for re-shaping the education future by facilitating innovative and sustainable approaches towards more value creation.

It was important to prioritize educational improvement as a public asset, preventing a generational catastrophe in the preparedness of the professionals and ensuring a sustainable educational resistance and recovery (for its resilience) [8]. Because of the COVID-19 restrictions, technologies used in online learning advanced and reached new levels never experienced before, so that EIs should take the advantages of that, despite the end or not of the COVID-19 restrictions.

Qatar's national education landscape serves as a significant foundation in online education in higher education for the research we are presenting. It is crucial to note that Qatar places significant importance on education, viewing it as a fundamental component of the country's development and a top national priority, as recognized by the General Secretariat for Development Planning in 2008. In the context of higher education institutions (HEIs), it becomes apparent that universities have excelled in providing robust support systems during emergency remote teaching. The university has demonstrated remarkable proficiency in delivering technical assistance. Drawing parallels with similar situations in universities across Australia, New Zealand, and Turkey, three distinct case scenarios reveal striking commonalities [9]. These scenarios emphasize the vital role of leadership initiatives essential for higher education transformation in times of a pandemic crisis.

In the context of the educational system in Qatar, the ability of educators and students to adapt and respond flexibly played a crucial role in ensuring the maintenance of a high-quality learning environment, even in the face of challenges. What stood out prominently in the Qatar case study was the rapid and effective responses demonstrated by EIs and educators when dealing with the constraints imposed by the COVID-19 pandemic [10–13]. Bensaid and Brahimi [14] provided a broad descriptive account of the state of digital education in Qatar during the pandemic by examining the diverse responses of EIs. Additionally, it outlined the measures taken by these institutions to adapt and thrive in this new environment. For instance, the Qatari Ministry of Education and Higher Education (MoEHE) launched a public service portal consolidating its services into a single platform. In the higher education sphere, Hamad Bin Khalifa University (HBKU) in Qatar maintained remote learning and ensured online access to audiobooks and books to support the continuity of the educational process.

To the best of our knowledge, there was no previous research focused on the digital transformation in higher education in Qatar during the COVID-19 pandemic. This study is dedicated to investigating the online education system within the context of crises such as the pandemic, specifically focusing on three selected universities in Qatar. We employed an online survey designed to target two distinct groups: educators and students. The primary objective of this research is to shed light on five key areas: (1) educational platform,

(2) digital tools and related support, (3) the learning environment and examination methods, (4) resilience levels during the transition to online education, and (5) perceptions of online classes in and beyond crises' occurrence. Through this comprehensive exploration, we aim to contribute valuable insights to the field of digital education during and beyond the pandemic era. This paper's structure is organized as follows: Section 2 highlights the relevant studies. Section 3 provides an overview of the empirical investigation methodology, offering a comprehensive view of the research approach employed. Section 4 elaborates on the data collection process, elucidating the sources, procedures, and techniques utilized. In Section 5, the focus turns to the empirical investigations, encompassing the presentation of findings and a thorough analysis aimed at deriving insightful outcomes. Section 6 extends this analytical perspective by examining the broader discussions within the field of study, emphasizing the research contributions made. Section 7 summarizes the paper's conclusions. Section 8 offers the encountered implications and limitations, as well as highlighting potential directions for future research endeavors in online education. For added convenience, the survey crafted for this study along with tabulated results are available in the Supplementary Materials.

## 2. Background

The emergence of the novel COVID-19 virus in late 2019 triggered a worldwide health crisis, swiftly declared a pandemic by the World Health Organization (WHO) in 2020. Its profound repercussions extended to various aspects of life, with a significant impact on the education field. To mitigate the virus's spread, many educational institutions quickly adopted remote learning and online teaching methods. This rapid shift to remote teaching during crises, referred to as "emergency remote teaching" (ERT), was coined by Hodges et al. (2020) [15] and subsequent scholars to ensure educational continuity. In this context, Sum and Oancea's (2022) [16] shed light on a prevalent issue in understanding the role of technology in remote teaching during the COVID-19 pandemic, as most reviewed studies revealed a deficiency in conceptualizing technology's role during this unprecedented time.

Highlighting the significance of students' autonomy and interest in online learning, Amelia et al. (2021) [17] discuss that such methods did not diminish students' curiosity or desire for knowledge. They stressed on the need for instructors to innovate and employ strategies that effectively engage students. Of note, student engagement in learning varies based on individual differences and preferences, including attitudes, mastery levels, skill transfer abilities, and information processing speed. Learners may also have distinct content presentation and interaction preferences. These individual traits and preferences significantly impact learning outcomes [18]. Therefore, it is crucial to acknowledge and accommodate these differences when designing online instruction. This can be achieved by incorporating diverse content delivery formats (such as animation, charts, presentations, and webinars), allowing for self-paced learning, promoting collaboration and peer teaching, and implementing varied assessment methods [19]. Indeed, the pivotal role of learners' motivation in facilitating an effective educational process has been a subject of extensive discourse and research, as highlighted in various studies, such as in [20,21].

Examining some countries experiences, Weber (2017) [22] recorded the initiation of the first e-learning initiatives in Qatar, notably among the U.S. satellite campuses within Education City. For example, Weill Cornell Medical College Qatar, founded in 2002, has integrated online and blended resources since its inception. In later stages, Hussain et al. (2020) [23] observed Qatar University's swift transition to high-quality distance learning, facilitated by the institution's long-standing and widespread use of foundational technological resources, underscoring the importance of institutional readiness in higher education. On the other hand, Heider (2021) [24] acknowledge effective implementations for remote education at Qatar University's Arabic for Non-Native Center. Nonetheless, challenges persist in verifying student identities during live streaming classes, exacerbated by insufficient connectivity quality for streaming access, among other factors.

Tang et al. (2021) [25] conducted a study in Hong Kong and reported that in order to enhance motivation among students, it is advisable to explore additional virtual activities. Moreover, fostering and promoting student-to-student interactions emerges as a valuable strategy worth considering. The European Commission (2020) [26] underscores the resurgence of the necessity to investigate opportunities for online teaching and learning due to the impact of COVID-19. In some European countries, it was observed that universities and students swiftly adjusted to the new changes, utilizing a combination of synchronous and asynchronous interaction and assessment methods at that time [27]. In the same context, the pandemic marked the commencement of a new era for Italian universities [2]. It has underscored that technological tools and distance learning not only enable doing more but also promote a newfound sense of sustainability. Recognizing that knowledge is the key to success, Italian universities must serve as the foundation for this renewal. In addition, it is increasingly evident that the emerged online and blended learning will continue to play a prominent role in Italian future educational delivery.

Additionally, Oliveira et al. (2021) [28] conducted a study in Portugal and Brazil, evidencing that the pandemic has left its mark on the utilization of remote education technologies, the educational journey, the adoption of information communication technology (ICT) platforms, and individual customized learning adjustments. While the use of ICT platforms proved to be highly advantageous, the personal adaptation process was predominantly difficult. However, a significant challenge has arisen due to the migration to online education as it has necessitated a mandatory shift in the perspectives of educational administrators, instructors, and learners regarding the importance of Emergency Remote Education (ERE) [29].

## 3. Methodology

This research aimed to source the experience and perceptions of the educators and students in three selected public universities in Qatar. The methodology employed in this study was designed to precisely capture the experiences and perceptions of educators and students, ensuring a direct correlation with the outlined research objectives, as detailed previously in Section 1. The survey (see the Supplementary Materials S1) was designed to target the educational environment during the pandemic time as an example of a crisis situation. This survey explored undergraduate and graduate students and educators at Hamad Bin Khalifa University (HBKU), Qatar University (QU), and the Community College of Qatar (CCQ). This work explored students' and educators' utilized support and the challenges experienced during the transformation in the education environment, as well as further suggesting new approaches to help assure the continuity of quality education in crisis events. The data collection tool in this study was Google Forms.

### 3.1. Ethical Consideration

This study obtained the necessary ethical approval from the ethics boards of the participating universities. The research adheres to principles such as voluntary participation, the right to withdraw at any point, and the confidentiality of participants' identities. HBKU serves as the primary host institution for this study, and the survey received ethical approval from the Hamad Bin Khalifa University—Institutional Review Board (HBKU-IRB) with the IRB Protocol Reference Number (QBRI-IRB-2024-4). Additionally, the Qatar University Institutional Review Board (QU-IRB) granted approval under reference number QU-IRB 1872-E/23.

### 3.2. Pilot Survey and Methodology Assessment

Prior to the main data collection phase, a meticulous survey design and pilot study were carried out to guarantee the reliability and validity of the survey structured components and to ensure that respondents' input could be accurately assessed. The survey design is driven by a comprehensive review of the existing literature [30]. The pilot study involved a sample of 30 students in the targeted universities, and they were asked to com-

plete the survey. The educator's survey was 90% similar to the student's survey; in other words, they were the same questions but from a different perspective: one as a learning perspective for the students and the other one as teaching perspective for the educators.

### 3.2.1. Reliability

In research that involves multi-item scales, the Cronbach's alpha test is a widely utilized method to assess the reliability of the scale. This test helps to determine whether all the items in the scale are effectively measuring the same underlying factor [31]. Additionally, to gauge the internal consistency of a survey instrument that incorporates multi-item scales, it is crucial to perform a reliability test, as emphasized by Churchill [32]. Reliability, in this context, refers to how accurately a test measures the intended construct. In this study, the Cronbach's alpha test, also known as the coefficient alpha, was employed to assess the reliability and internal consistency of the dataset. This test was used to ascertain whether the scales used in the study were dependable. Specifically, in this study, the sections of the student's survey exhibited reliability scores ranging from 0.720 to 0.895, confirming the robustness of this survey instrument for measuring the study's variables (Table S1). The reliability scores for various sections of the educator's survey ranged from 0.702 to 0.863, indicating strong reliability in measuring the study's variables (Table S6).

### 3.2.2. Validity

A validity coefficient is a gauge of how strong (or weak) that "usefulness" factor is; it provides the strength of that relationship between test results and criterion variables [33]. Tables S2–S5 in the Supplementary Materials display the inter-construct correlation validity for different sections of the student survey. In Table S2, which pertains to digital tools and related support, all correlation coefficients were significant with $p$-values less than 0.05 and a range from 0.372 to 0.598. Removing any items from this section would not increase the alpha beyond 0.720. Similarly, Table S3 focuses on the learning sphere and examination methods, with significant correlation coefficients ranging from 0.361 to 0.543. Deleting any of the nine items in this section would not increase the alpha beyond 0.731. Table S4 discusses resilience, showing significant correlation coefficients between 0.459 and 0.699. Removing any of the four items would not raise the alpha above 0.759. Finally, Table S5 examines online class perception, with significant correlation coefficients between 0.658 and 0.864, and deleting any of the four items would not increase the alpha beyond 0.895.

Tables S7–S10 in the Supplementary Materials present inter-construct correlation validity for the educator survey. In Table S7, related to digital tools and support, all correlations were significant (0.401 to 0.667), and removing items would not raise the alpha beyond 0.808. Table S8, on learning and examination methods, showed significant correlations (0.372 to 0.556), with no alpha increase above 0.702 if items were removed. Table S9, focused on resilience, had significant correlations (0.314 to 0.609), and deleting items would not raise the alpha beyond 0.710. Table S10, exploring online class perception, displayed significant correlations (0.701 to 0.769), with no alpha increase beyond 0.863 if items were removed.

## 4. Data Collection

We initiated the collection of responses through an online survey that we conducted. Data were acquired through a custom-designed online survey (see the Supplementary Materials), specifically tailored to address the unique circumstances brought about by the pandemic, focusing on three prominent HEIs in Qatar. One of these institutions is Qatar University (QU), which holds a significant position within Qatar's higher education landscape. Founded in 1977, QU has transformed into a comprehensive research-driven university, unwaveringly dedicated to achieving academic excellence, fostering innovation, and contributing to societal advancement. Another noteworthy institution is Hamad Bin Khalifa University (HBKU), which plays a pivotal role in Qatar's educational and research domain. HBKU offers a wide array of cutting-edge programs, encompassing both

undergraduate and graduate degrees, designed to address the evolving demands of the 21st-century workforce. The third institution, the Community College of Qatar (CCQ), contributes to educational diversity by providing a variety of associate and bachelor's degree programs. It aims to cater to a broad range of educational needs. The fact that these selected universities are large and highly diverse HEIs lends relevance to our findings, as they can serve as valuable insights for many other HEIs across the country. This study centers on two distinct groups in higher-education-level diploma, undergraduate, and graduate programs: students and educators hailing from three selected universities—QU, HBKU, and CCQ. The survey was distributed in August 2023 and subsequently reposted twice during September 2023. The survey was distributed comprehensively to encompass all students and educators within the three universities. This method ensured an inclusive representation across diverse educational levels and programs within the selected universities, aiming to capture a broad spectrum of perspectives from these prominent higher education institutions in Qatar. The sample size comprised 660 students and 103 educators, and the survey was closed after these data collection phases to commence the analysis. Google Forms was employed to program the survey, which gathered information on the demographics of students and educators, along with their experiences in online education during the teaching transformation. Each survey encompassed three sections: the first section collected socio-demographic information, the second explored details about the educational platform, and the third consisted of four components assessed using a five-point Likert scale (from total disagreement to total agreement). Overall, we believe that our selected sample of students and educators provides a reasonable and significantly representative view, considering our focus on three well-known universities in Qatar.

## 5. Empirical Analysis

In the quantitative data analysis phase, three stages were applied: data preparation; exploration; and analysis using SPSS Statistics for Windows, Version 29. SPSS was chosen in this study for its capabilities in managing data analysis and interpretation. It allows for the inclusion of a larger sample size, the creation of various charts and graphs, and the execution of significant statistical tests. Initially, data are input into SPSS and coded with unique identifiers. Subsequently, demographic profiles of respondents are presented using figures, charts, and tables. The upcoming section details the specific statistical tests applied to analyze the data.

Descriptive statistics represent the most prevalent form of statistical analysis within descriptive research. These statistics serve the purpose of summarizing and characterizing data, enabling the recognition of patterns and trends. They find utility in describing various types of data, including numerical, categorical, and ordinal data, while employing measures of central tendency. Within the scope of survey-based study, descriptive statistics prove invaluable in the examination of individual question responses. Furthermore, such statistics offer the capability to make comparisons between the responses of distinct respondent groups, facilitating, for instance, the examination of gender-based variations.

### 5.1. Socio-Demographic Data

5.1.1. Students

Survey results were obtained from a sample comprising 660 students, drawn from HBKU, QU, and CCQ. As illustrated in Table 1, a predominant proportion of the participants, totaling 70.9%, were females. The age distribution of students revealed that the largest group fell within the 18–22 age range (47.1%), followed by those aged 22–27 (23.87%). A smaller portion of students were aged 32 and older (15.9%), while the 27–32 age group represented the smallest percentage (13.2%). Moreover, QU had the highest representation among the surveyed students, accounting for 67.7% of the respondents, followed by CCQ at 24.2%, and HBKU at 8.0%. These findings provide insights into the demographics of the surveyed student population.

**Table 1.** Characteristics of the student's participants (*n* = 660).

| Characteristics | | *n* | % |
|---|---|---|---|
| Gender | Male | 192 | 29.1 |
| | Female | 468 | 70.9 |
| Age | 18 to less than 22 | 311 | 47.1 |
| | 22 to less than 27 | 157 | 23.8 |
| | 27 to less than 32 | 87 | 13.2 |
| | 32+ | 105 | 15.9 |
| University | HBKU | 53 | 8 |
| | QU | 447 | 67.7 |
| | CCQ | 160 | 24.2 |

Table 2 highlights the characteristics of academic information. The first year was the most common academic year among respondents, accounting for 33.6% of the students. This was followed by students in their fourth year or beyond at 24.8%, the second year at 22.4%, and the third year at 19.1%. Also, Table 2 shows the visual representation of these groups. In terms of education levels, the majority of respondents held a bachelor's degree (65.0%), followed by diploma holders (20.8%), master's degree recipients (7.7%), and Ph.D. degree holders (6.5%), Finally, the most common colleges among respondents were the College of Engineering (18.8%), followed by the College of Arts and Science (17.1%), the College of Humanities and Social Sciences (11.8%), and the College of Business and Economics (10.8%). The study sample exhibited diversity across various demographics, including gender, age, university affiliation, academic year, educational background, and college enrollment.

**Table 2.** Academic information of the student's participations.

| Characteristics | | *n* | % |
|---|---|---|---|
| Year of studying | First year | 222 | 33.6 |
| | Second year | 148 | 22.4 |
| | Third year | 126 | 19.1 |
| | Fourth year+ | 164 | 24.8 |
| Education level | Bachelor level | 429 | 65 |
| | Diploma * | 137 | 20.8 |
| | Graduate master's level | 51 | 7.7 |
| | Graduate Ph.D. level | 43 | 6.5 |
| College | College of Engineering | 124 | 18.8 |
| | College of Arts and Science | 113 | 17.1 |
| | College of Humanities and Social Sciences | 78 | 11.8 |
| | College of Business and Economics | 71 | 10.8 |
| | College of Education | 46 | 7 |
| | College of Health and Life Sciences | 41 | 6.2 |
| | College of Sharia and Islamic Studies | 36 | 5.5 |
| | College of Law | 21 | 3.2 |
| | College of Medicine and Pharmacy | 19 | 2.9 |
| | College of Computer Science | 13 | 2 |
| | Other | 98 | 14.8 |

* Diploma is a qualification awarded upon completion of a specific program, such as those offered by Qatar University and the Community College of Qatar. These programs emphasize practical skills and are shorter in duration than traditional bachelor's degrees, preparing students for entry-level positions in various fields.

### 5.1.2. Educators

The findings of the survey involving 103 educators from HBKU, QU, and CCQ revealed that the sample consisted of 72 male educators and 31 female educators, accounting for 69.9% and 30.1%, respectively, as shown in Table 3. Most educators in the sample fell within the age group of 40 to 49 (39.8%), with the second-largest group aged between 50 and 59

(29.1%). The smallest percentage of educators in the sample belonged to the age group of 25 to 29 (4.9%), as depicted in Table 3. In the educational landscape of our study, QU took center stage as the most prominently represented institution (68%), with the CCQ (17.5%) and HBKU (14.6%) following closely behind.

**Table 3.** Characteristics of the educator's participants (*n* = 103).

| Characteristics | | *n* | % |
|---|---|---|---|
| Gender | Male | 72 | 69.9 |
| | Female | 31 | 30.1 |
| Age | 25 to less than 30 | 5 | 4.9 |
| | 30 to less than 40 | 20 | 19.4 |
| | 40 to less than 50 | 41 | 39.8 |
| | 50 to less than 60 | 30 | 29.1 |
| | 60+ | 7 | 6.8 |
| University | HBKU | 15 | 14.6 |
| | QU | 70 | 68 |
| | CCQ | 18 | 17.5 |

When considering the education levels at which educators teach, it is evident that the majority of educators in the sample were involved in teaching at least at a bachelor's degree level (32.0%). Graduate-level qualifications, encompassing both master's and Ph.D. degrees, were the educational domain for 26.2% of the sample. Furthermore, 11.7% were engaged in teaching at the master's degree level, while 10.7% were involved in teaching at the Ph.D. level. Moreover, 17.5% of educators were engaged in teaching across multiple higher educational levels. There was a smaller proportion of educators (1.9%) in the sample who were involved in teaching at the diploma level, as depicted in Table 4. To provide a complete picture, it is worth noting that the most common colleges in the sample were the College of Engineering and the College of Business and Economics, both with a representation of 13.6%. Additionally, educators in the sample were affiliated with a variety of other colleges, encompassing the College of Arts and Science, the College of Humanities and Social Sciences, the College of Education, the College of Health and Life Sciences, the College of Medicine and Pharmacy, the College of Law, and the College of Computer Science. This diversity signifies a broad spectrum of teaching disciplines within the sample.

**Table 4.** Academic information of the educator's participations.

| Characteristics | | *n* | % |
|---|---|---|---|
| Education level | Diploma | 2 | 1.9 |
| | Bachelor level | 33 | 32.0 |
| | Graduate Level (master's) | 12 | 11.7 |
| | Graduate Level (Ph.D.) | 11 | 10.7 |
| | Graduate Level (both master's and Ph.D.) | 27 | 26.2 |
| | More than one level | 18 | 17.5 |
| College | College of Engineering | 12 | 11.7 |
| | College of Arts and Science | 12 | 11.7 |
| | College of Humanities and Social Sciences | 14 | 13.6 |
| | College of Business and Economics | 14 | 13.6 |
| | College of Education | 12 | 11.7 |
| | College of Health and Life Sciences | 6 | 5.8 |
| | College of Medicine and Pharmacy | 8 | 7.8 |
| | College of Law | 10 | 9.7 |
| | College of Computer Science | 6 | 5.8 |
| | Other | 9 | 8.7 |

*5.2. Educational Platform*

5.2.1. Students

Table 5 presents a comprehensive overview of the study sample's characteristics concerning their engagement with educational platforms. An interesting observation emerged as 65.5% of the students were at the time not engaged as full-time employees. This suggests that a significant portion of the sample was primarily focused on their studies. Conversely, 34.5% were at the time employed full-time, which indicates that this dual commitment to work and education may have implications for their academic workload and preferences. When examining their preferences for post-pandemic learning modalities, it was notable that 44.4% of the students expressed a preference for a blend of virtual and in-person classes. This indicates a growing acceptance of hybrid learning approaches, aligning with the changing educational landscape. Moreover, a substantial proportion of the students, approximately 39.4%, had previous exposure to online classes before the pandemic. This prior experience may have influenced their preferences and readiness to embrace online learning. A noteworthy trend was that 48.6% of the students favored hybrid instruction post-pandemic, which incorporates both online and face-to-face elements. This underscores the demand for flexible and adaptable learning options and models. In the realm of educational technology, a significant 62.3% of the students commonly employed Microsoft Teams for their online educational needs. This widespread usage highlights the platform's popularity and potentially its impact on the students' virtual learning experiences. Furthermore, 38.3% of the students exhibited awareness of immersive technologies and the metaverse. This is a notable finding, indicating a degree of technological literacy within the sample and a potential readiness to explore innovative learning tools.

**Table 5.** The characteristics of the study sample towards educational platform (Students).

| Variables | Options | *n* | % |
|---|---|---|---|
| Are you currently a full-time employee? | No | 432 | 65.5 |
| | Yes | 228 | 34.5 |
| What type of attendance do you prefer after the pandemic? | Virtually (online class) | 103 | 15.6 |
| | Physically (in-person) | 264 | 40.0 |
| | Combination of both | 293 | 44.4 |
| Have you done any online classes before the pandemic? | No | 400 | 60.6 |
| | Yes | 260 | 39.4 |
| How do you prefer the learning process to be after the pandemic? | Face-to-face (class sessions take place 100% in the classroom) | 255 | 38.6 |
| | Hybrid (online and face-to-face instruction are integrated) | 321 | 48.6 |
| | Online (all instruction, interaction, and activities take place online) | 84 | 12.7 |
| What was (or still is) the platform most used by you in online education? | Microsoft Teams | 411 | 62.3 |
| | Zoom app | 103 | 15.6 |
| | Google Meet | 33 | 5.0 |
| | WebEx | 40 | 6.1 |
| | Other | 73 | 11.1 |
| Have you already heard about immersive technologies and metaverse? | No | 407 | 61.7 |
| | Yes | 253 | 38.3 |

In general, based on the survey results in Table 5, it was evident that higher education students have varied preferences for their learning environment. A minority preferred exclusively online classes, indicating some level of comfort or convenience with virtual learning. However, the majority either favored in-person classes, emphasizing the value they place on direct, physical interaction in an educational setting, or a combination of both online and in-person, suggesting a desire for a flexible and hybrid learning approach. This diversity in preferences highlights the importance of offering multiple modes of education to cater to different learning styles and needs.

The data presented in Table 5 portray a dynamic educational landscape, where students exhibited diverse preferences and technological fluency. The prominence of hybrid learning and the increasing familiarity with emerging technologies suggest an evolving educational paradigm that warrants further investigation and adaptation. The concept of the metaverse was added to the survey since it has gained significant attention in recent years. And it refers to advancements in technology and the desire to create immersive digital experiences.

### 5.2.2. Educators

The educators exhibited several noteworthy trends. Firstly, a significant proportion of educators (66.1%) possessed over a decade of teaching experience, indicating a wealth of knowledge and expertise within the sample. Furthermore, a preference for online classes following the pandemic was prevalent among the majority of educators (55.3%), reflecting a desire for traditional in-person instruction. Moreover, a substantial segment of educators (57.3%) had prior experience delivering online classes, seminars, or conferences before the pandemic, highlighting a degree of familiarity with virtual teaching modalities. Notably, a preference for hybrid learning post-pandemic was endorsed by the majority of educators (49.5%), signifying an acknowledgment of the value of blended instructional approaches. When examining the digital landscape, Microsoft Teams emerged as the dominant educational platform among educators in the sample (49.5%). This substantial usage implies the platform's widespread adoption and its potential impact on the virtual teaching experience. In addition, a significant proportion of the educator's cohort (59.2%) displayed a tech-savvy awareness of immersive technologies and the metaverse, underlining their technological literacy. This recognition may signify a readiness to explore innovative tools and models in the realm of education. Table 6 provides a comprehensive summary of the records and preferences obtained within the educators' sample and the findings stated that educators showed that opinions were almost evenly split between preference for face-to-face instruction (48.5%) and hybrid teaching (49.5%), which integrates both online and in-person elements. Only a very small percentage (1.9%) preferred exclusively online instruction. This indicates a strong inclination towards maintaining personal interaction in the educational process, either through traditional classroom settings or a blend of online and offline methods. The minimal preference for fully online instruction suggests that educators see significant value in in-person or mixed teaching approaches post-pandemic. Further, the potential cause for this preference might be attributed to the abrupt and challenging transition to online learning. Educators have reported that the shift in the mode of education was far from seamless, hindered by inadequate infrastructure and a deficit in technological proficiency within certain educational institutions.

### *5.3. Digital Tools and Related Support*
### 5.3.1. Students

Table 7 presents a detailed evaluation of the participants' perceptions regarding digital tools and related support and shows the distribution of the participants' perceptions regarding digital tools and related support in an educational context. The data revealed a high level of agreement among participants, as evidenced by an overall mean score of ($3.79 \pm 0.677$), falling within the range of (3.40 to less than 4.20). This finding underscores the participants' general satisfaction with the digital tools and support they have access to, suggesting that these resources significantly contribute to their learning experience. Among the seven items evaluated, six received high mean scores, while one attained a medium-level score. The highest-rated item, "I was able to access the online material quickly," received a high mean score of ($4.14 \pm 0.805$). This indicates that participants consistently found it easy to access the necessary online learning materials promptly, reflecting the effectiveness of the digital tools in providing seamless access to resources. Conversely, the item "I received sufficient training to use the online platform" obtained a medium mean score of ($3.35 \pm 1.161$), signifying the lowest-rated aspect among the evaluated items. This

suggests that there may be room for improvement in terms of providing adequate training to enhance participants' proficiency in using the online platform effectively. It is essential to address this aspect to ensure that participants can leverage the full potential of the online learning environment.

**Table 6.** The characteristics of the study sample towards educational platforms (Educators).

| Variables | Options | *n* | % |
|---|---|---|---|
| Experience | Less than 5 years | 16 | 15.5 |
| | 5 to less than 10 years | 19 | 18.4 |
| | 10 to less than 15 years | 22 | 21.4 |
| | 15 to less than 20 years | 18 | 17.5 |
| | 20 years+ | 28 | 27.2 |
| What type of attendance do you prefer after the pandemic? | Virtually (online class) | 57 | 55.3 |
| | Physically (in-person) | 2 | 1.9 |
| | Combination of both | 44 | 42.7 |
| Did you give any online classes, seminars, or conferences before the pandemic? | No | 44 | 42.7 |
| | Yes | 59 | 57.3 |
| How do you prefer the learning process to be after the pandemic? | Face-to-face (class sessions take place 100% in the classroom) | 50 | 48.5 |
| | Hybrid (online and face-to-face instruction are integrated) | 51 | 49.5 |
| | Online (all instruction, interaction, and activities take place online) | 2 | 1.9 |
| What is the platform you use most in online classes? | Microsoft Teams | 51 | 49.5 |
| | Zoom app | 5 | 4.9 |
| | Google Meet | 2 | 1.9 |
| | WebEx | 18 | 17.5 |
| | Other | 27 | 26.2 |
| Have you already heard about immersive technologies and the metaverse? | No | 42 | 40.8 |
| | Yes | 61 | 59.2 |

**Table 7.** Descriptive statistics for participants' responses towards digital tools and related support.

| Question | | 1 | 2 | 3 | 4 | 5 | Mean | SD | Level | Rank |
|---|---|---|---|---|---|---|---|---|---|---|
| The e-learning platform is easy to use | n | 5 | 15 | 94 | 337 | 209 | 4.11 | 0.780 | High | 2 |
| | % | 0.8 | 2.3 | 14.2 | 51.1 | 31.7 | | | | |
| I was able to access the online material quickly | n | 4 | 24 | 77 | 323 | 232 | 4.14 | 0.805 | High | 1 |
| | % | 0.6 | 3.6 | 11.7 | 48.9 | 35.2 | | | | |
| I could attend the online lecture without any interruption | n | 22 | 86 | 130 | 271 | 151 | 3.67 | 1.068 | High | 5 |
| | % | 3.3 | 13.0 | 19.7 | 41.1 | 22.9 | | | | |
| I received sufficient training to use the online platform | n | 38 | 138 | 156 | 211 | 117 | 3.35 | 1.161 | Medium | 7 |
| | % | 5.8 | 20.9 | 23.6 | 32.0 | 17.7 | | | | |
| The college offered sufficient technical support for the used platform | n | 13 | 52 | 178 | 264 | 153 | 3.75 | 0.965 | High | 4 |
| | % | 2.0 | 7.9 | 27.0 | 40.0 | 23.2 | | | | |
| I am satisfied with the technology and software I used in online learning | n | 14 | 33 | 89 | 324 | 200 | 4.00 | 0.911 | High | 3 |
| | % | 2.1 | 5.0 | 13.5 | 49.1 | 30.3 | | | | |
| I want online classes to be part of the learning process after the pandemic | n | 76 | 94 | 92 | 206 | 192 | 3.52 | 1.345 | High | 6 |
| | % | 11.5 | 14.2 | 13.9 | 31.2 | 29.1 | | | | |
| Digital tools and related support | | | | | | | 3.79 | 0.677 | High | |

Strongly disagree = 1, disagree = 2, nether = 3, agree = 4, strongly agree = 5.

#### 5.3.2. Educators

Similarity, Table 8 offers a detailed overview of how educators perceive digital tools and the support associated with them in an educational setting and shows educators' perceptions regarding digital tools and related support in the educational context. The data showcased a remarkable level of agreement among educators, with an overall mean score of (3.83 ± 0.670), falling within the range of (3.40 to less than 4.20). This finding underscores the educators' high satisfaction with the digital tools and support they have access to, highlighting the pivotal role these resources play in their educational endeavors. Among the seven items assessed, one item garnered a very high mean score, five items received high scores, and one item achieved a medium-level score. The highest-rated item, "The education institution offered sufficient technical support," obtained an exceptionally high mean score of (4.36 ± 0.684). This suggests that educators deeply value the technical support provided by their institutions, signifying its crucial role in enabling educators to utilize digital tools effectively. Conversely, the item "I wish online classes were part of the learning process after the pandemic" received a medium mean score of (3.23 ± 1.254), representing the lowest-rated aspect among the evaluated items. This observation emphasizes the challenges posed by online learning, experienced by both educators and students. Nevertheless, it is essential to acknowledge that online learning is a valuable pedagogical tool with an increasingly vital role in education's future landscape. Embracing and enhancing online learning methods may be the key to delivering quality education in a rapidly evolving educational landscape.

**Table 8.** Descriptive statistics for educators' responses towards digital tools and related support.

| Question | | 1 | 2 | 3 | 4 | 5 | Mean | SD | Level | Rank |
|---|---|---|---|---|---|---|---|---|---|---|
| E-learning platform improves the learning processes | n | 1 | 13 | 29 | 33 | 27 | 3.70 | 1.027 | High | 6 |
| | % | 1.0 | 12.6 | 28.2 | 32.0 | 26.2 | | | | |
| I am satisfied with the used platform | n | 1 | 5 | 20 | 55 | 22 | 3.89 | 0.827 | High | 3 |
| | % | 1.0 | 4.9 | 19.4 | 53.4 | 21.4 | | | | |
| I could deliver online lectures without any interruption | n | 2 | 8 | 17 | 52 | 24 | 3.85 | 0.933 | High | 4 |
| | % | 1.9 | 7.8 | 16.5 | 50.5 | 23.3 | | | | |
| The education institution offered sufficient technical support | n | - | 2 | 6 | 48 | 47 | 4.36 | 0.684 | Very high | 1 |
| | % | - | 1.9 | 5.8 | 46.6 | 45.6 | | | | |
| I received sufficient training to use the e-learning platform | n | - | 11 | 11 | 50 | 31 | 3.98 | 0.918 | High | 2 |
| | % | - | 10.7 | 10.7 | 48.5 | 30.1 | | | | |
| I wish for online classes as part of the learning process after the pandemic | n | 13 | 16 | 24 | 34 | 16 | 3.23 | 1.254 | Medium | 7 |
| | % | 12.6 | 15.5 | 23.3 | 33.0 | 15.5 | | | | |
| Technologies helped in the teaching and learning process and improved educational competency during the pandemic | n | 5 | 10 | 17 | 39 | 32 | 3.81 | 1.129 | High | 5 |
| | % | 4.9 | 9.7 | 16.5 | 37.9 | 31.1 | | | | |
| Digital tools and related support | | | | | | | 3.83 | 0.670 | High | |

Strongly disagree = 1, disagree = 2, nether = 3, agree = 4, strongly agree = 5.

#### 5.4. Learning Material and Examination Methods

##### 5.4.1. Students

Table 9 offers valuable insights into the participants' perceptions regarding learning materials and examination methods. The overall mean score of (3.35 ± 0.654), falling within the interval of (2.60 to less than 3.40), indicates that participants expressed a medium level of agreement in this regard. An interesting trend emerged when we delved deeper. Five out of the seven items achieved a high level of agreement, while three items garnered a medium level, and a single item achieved a low level of agreement. Notably, the item that ranked highest, "E-learning contents are varied, including video lessons, worksheets,

textbooks, and assessments," received a high mean score of (3.94 ± 0.926). This suggests that participants were generally content with the diverse array of learning materials at their disposal. Conversely, the item that ranked the lowest was the reversed statement, "I lose my focus during online classes," which obtained a low mean score of (2.43 ± 1.248). This indicates that only a minority of participants managed to maintain their focus during online classes. Notably, 7.7% strongly disagreed with this item, and 13.9% simply disagreed with it.

**Table 9.** Descriptive statistics for participants' responses towards learning material and examination methods.

| Question | | 1 | 2 | 3 | 4 | 5 | Mean | SD | Level | Rank |
|---|---|---|---|---|---|---|---|---|---|---|
| The online materials were ready in time during the emergency transition | n | 20 | 45 | 146 | 285 | 164 | 3.80 | 0.988 | High | 3 |
| | % | 3.0 | 6.8 | 22.1 | 43.2 | 24.8 | | | | |
| The online materials provided were comprehensive | n | 11 | 29 | 151 | 331 | 138 | 3.84 | 0.860 | High | 2 |
| | % | 1.7 | 4.4 | 22.9 | 50.2 | 20.9 | | | | |
| I am satisfied with the online examination method during online education | n | 45 | 81 | 139 | 236 | 159 | 3.58 | 1.176 | High | 5 |
| | % | 6.8 | 12.3 | 21.1 | 35.8 | 24.1 | | | | |
| Online (live) class are more effective than recorded classes | n | 43 | 76 | 138 | 192 | 211 | 3.68 | 1.216 | High | 4 |
| | % | 6.5 | 11.5 | 20.9 | 29.1 | 32.0 | | | | |
| E-learning contents are varied including video lessons, worksheets, textbooks, and assessments | n | 13 | 38 | 112 | 310 | 187 | 3.94 | 0.926 | High | 1 |
| | % | 2.0 | 5.8 | 17.0 | 47.0 | 28.3 | | | | |
| I lose my focus during the online classes (reversed item) * | n | 51 | 92 | 137 | 193 | 187 | 2.43 | 1.248 | Low | 9 |
| | % | 7.7 | 13.9 | 20.8 | 29.2 | 28.3 | | | | |
| I feel exhausted related to online classes (reversed item) | n | 82 | 142 | 152 | 167 | 117 | 2.86 | 1.287 | Medium | 8 |
| | % | 12.4 | 21.5 | 23.0 | 25.3 | 17.7 | | | | |
| I do not have a favourable environment to study at home (reversed item) | n | 103 | 170 | 152 | 159 | 76 | 3.10 | 1.255 | Medium | 6 |
| | % | 15.6 | 25.8 | 23.0 | 24.1 | 11.5 | | | | |
| If I were to select between online and presential learning, I would choose online | n | 138 | 150 | 140 | 124 | 108 | 2.87 | 1.375 | Medium | 7 |
| | % | 20.9 | 22.7 | 21.2 | 18.8 | 16.4 | | | | |
| Learning material and examination methods | | | | | | | 3.35 | 0.654 | Medium | |

Strongly disagree = 1, disagree = 2, nether = 3, agree = 4, strongly agree = 5. * Reversed item = five-point Likert scale (from total agreement to total disagreement).

### 5.4.2. Educators

Table 10 provides an overview of educators' responses regarding learning materials and examination methods. Educators expressed a medium level of agreement, indicated by an overall mean score of (3.21 ± 0.634), within the interval of (2.60 to less than 3.40). With a closer scrutiny, four out of seven items garnered a medium level of agreement, two items received high agreement, and one item registered a low level of agreement. The item with the highest rank was "Despite the challenges of remote teaching, educators can teach students effectively," with a notably high mean score of (3.92 ± 0.788). This observation suggests educators' dedication to their students and their capacity to deliver effective teaching even in challenging circumstances, reflecting positively on their ability to navigate changes. In contrast, the item ranking lowest was the statement "Educators are satisfied with the online examination method," obtaining a low mean score of (2.59 ± 1.240). This finding implies that educators hold concerns regarding the fairness and effectiveness of online examination methods. Such apprehensions are valid, given the challenges associated with proctoring online examinations and the potential limitations of online assessments in accurately evaluating student learning.

**Table 10.** Descriptive statistics for educators' responses towards learning material and examination methods.

| Question | | 1 | 2 | 3 | 4 | 5 | Mean | SD | Level | Rank |
|---|---|---|---|---|---|---|---|---|---|---|
| I have not needed to provide any new learning material for the online-based class | N | 6 | 39 | 19 | 32 | 7 | 2.95 | 1.097 | Medium | 5 |
| | % | 5.8 | 37.9 | 18.4 | 31.1 | 6.8 | | | | |
| The online materials provided are comprehensive | N | 1 | 11 | 25 | 49 | 17 | 3.68 | 0.910 | High | 2 |
| | % | 1.0 | 10.7 | 24.3 | 47.6 | 16.5 | | | | |
| I am satisfied with the online examination method | N | 24 | 29 | 22 | 21 | 7 | 2.59 | 1.240 | Low | 7 |
| | % | 23.3 | 28.2 | 21.4 | 20.4 | 6.8 | | | | |
| Despite the challenges of remote teaching, I was able to teach my students effectively | N | 1 | 6 | 12 | 65 | 19 | 3.92 | 0.788 | High | 1 |
| | % | 1.0 | 5.8 | 11.7 | 63.1 | 18.4 | | | | |
| I was able to understand students' thoughts, desires, and fears | N | 5 | 15 | 30 | 45 | 8 | 3.35 | 0.987 | Medium | 3 |
| | % | 4.9 | 14.6 | 29.1 | 43.7 | 7.8 | | | | |
| I am satisfied with online discussions with the students during the lectures | N | 6 | 25 | 17 | 47 | 8 | 3.25 | 1.091 | Medium | 4 |
| | % | 5.8 | 24.3 | 16.5 | 45.6 | 7.8 | | | | |
| I spent less time in online-based classes than in face-to-face classes | N | 16 | 36 | 17 | 30 | 4 | 2.71 | 1.160 | Medium | 6 |
| | % | 15.5 | 35.0 | 16.5 | 29.1 | 3.9 | | | | |
| Learning material and examination methods | | | | | | | 3.21 | 0.634 | Medium | |

Strongly disagree = 1, disagree = 2, nether = 3, agree = 4, strongly agree = 5.

*5.5. Resilience*

5.5.1. Students

Table 11 provides a comprehensive view of students' responses concerning their resilience in adapting to online education. The collective disposition towards resilience is reflected in an overall mean score of (3.45 ± 0.839), which positions the responses within the high agreement range, defined as (3.40 to less than 4.20). Upon closer examination, it became evident that two of the four items under scrutiny garnered high levels of agreement. These items indicated a notable level of resilience among participants in the face of the challenges posed by online education. Notably, the item "I was capable of adjusting to online education" secured the highest mean score at (3.74 ± 1.019), suggesting that participants demonstrated a remarkable capacity for adaptation, enabling them to effectively navigate the new online learning environment. Conversely, the remaining two items in this category attained only a moderate level of agreement. The item "I am satisfied with teamwork engagement while studying online" received the lowest mean score (3.18 ± 1.220), signaling potential difficulties participants encountered when attempting to engage in collaborative efforts with their peers in the online learning setting.

5.5.2. Educators

Table 12 provides an analysis of the resilience demonstrated by educators in the face of new developments, particularly the challenges brought about by the pandemic. The educators' collective disposition towards resilience was marked by an overall mean score of (4.01 ± 0.523), placing their responses within the high agreement range (defined as scores falling between 3.40 and less than 4.20). Notably, all items within this category obtained high levels of agreement, underscoring the educators' remarkable resilience during these transformative times. The item that received the highest mean score was "I adapted quickly to new developments" (4.17 ± 0.706). This outcome signifies that educators displayed a remarkable ability to acquire new skills and swiftly adjust their teaching methods to meet the evolving needs of their students, reflecting their adaptability in the face of changing circumstances. In contrast, the item "I could tolerate high levels of

ambiguity and uncertainty about situations" attained the lowest rank among the items, but with a high mean score (3.78 ± 0.851). This finding suggests that educators may have encountered challenges in managing the ambiguity and uncertainty arising from the pandemic. Understandably, the pandemic represented a novel and daunting experience for educators and society at large. Nevertheless, it is vital to recognize that ambiguity and uncertainty are inherent aspects of life, and educators must cultivate the capacity to effectively address these challenges to maintain their effectiveness as teachers.

**Table 11.** Descriptive statistics for student's responses towards resilience.

| Question | | 1 | 2 | 3 | 4 | 5 | Mean | SD | Level | Rank |
|---|---|---|---|---|---|---|---|---|---|---|
| I am satisfied with the online discussion with my teachers | n | 39 | 79 | 152 | 252 | 138 | 3.56 | 1.123 | High | 2 |
| | % | 5.9 | 12.0 | 23.0 | 38.2 | 20.9 | | | | |
| I spent less time in studying during online education | n | 55 | 118 | 159 | 222 | 106 | 3.31 | 1.181 | Medium | 3 |
| | % | 8.3 | 17.9 | 24.1 | 33.6 | 16.1 | | | | |
| I am satisfied with teamwork engagement while studying online | n | 66 | 139 | 167 | 184 | 104 | 3.18 | 1.220 | Medium | 4 |
| | % | 10.0 | 21.1 | 25.3 | 27.9 | 15.8 | | | | |
| I was capable to adjust to online education | n | 31 | 43 | 136 | 308 | 142 | 3.74 | 1.019 | High | 1 |
| | % | 4.7 | 6.5 | 20.6 | 46.7 | 21.5 | | | | |
| Resilience | | | | | | | 3.45 | 0.839 | High | |

Strongly disagree = 1, disagree = 2, nether = 3, agree = 4, strongly agree = 5.

**Table 12.** Descriptive statistics for educators' responses towards resilience.

| | | 1 | 2 | 3 | 4 | 5 | Mean | SD | Level | Rank |
|---|---|---|---|---|---|---|---|---|---|---|
| I was able to adapt my teaching to the conditions of online teaching while upholding my usual standards of quality | n | - | 4 | 14 | 67 | 18 | 3.96 | 0.685 | High | 3 |
| | % | - | 3.9 | 13.6 | 65.0 | 17.5 | | | | |
| I adapted quickly to new developments | n | - | 3 | 9 | 58 | 33 | 4.17 | 0.706 | High | 1 |
| | % | - | 2.9 | 8.7 | 56.3 | 32.0 | | | | |
| I could convert challenges into opportunities and build up my experiences | n | - | 2 | 10 | 64 | 27 | 4.13 | 0.652 | High | 2 |
| | % | - | 1.9 | 9.7 | 62.1 | 26.2 | | | | |
| I could tolerate high levels of ambiguity and uncertainty about situations | n | - | 9 | 24 | 51 | 19 | 3.78 | 0.851 | High | 4 |
| | % | - | 8.7 | 23.3 | 49.5 | 18.4 | | | | |
| Resilience | | | | | | | 4.01 | 0.523 | High | |

Strongly disagree = 1, disagree = 2, nether = 3, agree = 4, strongly agree = 5.

### 5.6. Online Classes Perception

5.6.1. Students

Table 13 reveals participants' perceptions of online classes, indicating a high level of agreement with an overall mean score of (3.36 ± 1.017) within the interval of (2.60 to less than 3.40). Out of the four items, two received high-level agreement, while the remaining two demonstrated a medium level of agreement. Notably, the item ranked highest, "I believe online classes will aid the learning process during and post the pandemic," received a high mean score of (3.73 ± 1.173). Conversely, the lowest-ranked item, "I learn better through online classes," received a low mean score of (2.91 ± 1.304). This suggests a varying degree of belief in the efficacy of online classes within the sample, with the majority expressing positive views, although a portion remains skeptical.

**Table 13.** Descriptive statistics for students' responses towards online class perception.

| | | 1 | 2 | 3 | 4 | 5 | Mean | SD | Level | Rank |
|---|---|---|---|---|---|---|---|---|---|---|
| I find online classes interesting and useful | n | 45 | 74 | 161 | 251 | 129 | 3.52 | 1.129 | High | 2 |
| | % | 6.8 | 11.2 | 24.4 | 38.0 | 19.5 | | | | |
| I am satisfied with the online interactions with my classmates | n | 62 | 125 | 153 | 215 | 105 | 3.27 | 1.208 | Medium | 3 |
| | % | 9.4 | 18.9 | 23.2 | 32.6 | 15.9 | | | | |
| I learn better through online classes | n | 119 | 140 | 174 | 133 | 94 | 2.91 | 1.304 | Medium | 4 |
| | % | 18.0 | 21.2 | 26.4 | 20.2 | 14.2 | | | | |
| I believe online classes will aid the learning process during and after the pandemic | n | 48 | 54 | 116 | 252 | 190 | 3.73 | 1.173 | High | 1 |
| | % | 7.3 | 8.2 | 17.6 | 38.2 | 28.8 | | | | |
| Online classes perception | | | | | | | 3.36 | 1.017 | Medium | |

Strongly disagree = 1, disagree = 2, nether = 3, agree = 4, strongly agree = 5.

### 5.6.2. Educators

Table 14 provides a detailed insight into these perceptions, highlighting a medium-level agreement with an overall mean score of (3.09 ± 0.901), falling within the interval of (2.60 to less than 3.40). Within this evaluation, two out of the four items received a high level of agreement, one demonstrated medium-level agreement, and the remaining item registered a low level of agreement. Remarkably, the highest-ranked item, "I find online classes interesting and useful," garnered a high mean score of (3.50 ± 0.969), indicating that educators recognize the potential benefits associated with online classes. However, the observation that "I believe the students learn better through online classes" received the lowest ranking, with a mean score of (2.37 ± 1.010), suggesting that educators harbor concerns about the efficacy of online learning. This discrepancy underscores the nuanced attitudes of educators towards online education, recognizing its potential while remaining skeptical of its effectiveness.

**Table 14.** Descriptive statistics for educators' responses towards online class perception.

| Question | | 1 | 2 | 3 | 4 | 5 | Mean | SD | Level | Rank |
|---|---|---|---|---|---|---|---|---|---|---|
| I find online classes interesting and useful | n | 4 | 12 | 26 | 50 | 11 | 3.50 | 0.969 | High | 1 |
| | % | 3.9 | 11.7 | 25.2 | 48.5 | 10.7 | | | | |
| I am satisfied with the online interactions with my students | n | 12 | 25 | 16 | 44 | 6 | 3.07 | 1.174 | Medium | 3 |
| | % | 11.7 | 24.3 | 15.5 | 42.7 | 5.8 | | | | |
| I believe the students learn better through online classes | n | 24 | 31 | 36 | 10 | 2 | 2.37 | 1.010 | Low | 4 |
| | % | 23.3 | 30.1 | 35.0 | 9.7 | 1.9 | | | | |
| I believe online classes will aid the learning process during and after the pandemic | n | 6 | 14 | 28 | 41 | 14 | 3.42 | 1.071 | High | 2 |
| | % | 5.8 | 13.6 | 27.2 | 39.8 | 13.6 | | | | |
| Online classes perception | | | | | | | 3.09 | 0.901 | Medium | |

Strongly disagree = 1, disagree = 2, nether = 3, agree = 4, strongly agree = 5.

## 6. Discussion

To the best of our knowledge, this research marks the initial endeavor to examine online education within the higher education level in Qatar. It casts illumination upon five primary aspects that this paper concentrates on: the educational platform, digital tools and the corresponding support mechanisms, the learning environment and assessment methodologies, the level of adaptability exhibited during the shift to online education, and the perceptions of online classes among both students and educators in the context of the pandemic-driven transition to online education.

### 6.1. Digital Tools and Related Support

Based on the aforementioned analysis of responses from both students and educators, it was established that there exists a notable correlation between the features of the utilized platform and the degree of contentment regarding digital tools and the accompanying support. This suggests that students generally expressed satisfaction with the digital tools and support they had access to, finding them beneficial for their learning. Additionally, educators exhibited a high level of satisfaction with the digital tools and support, acknowledging their positive impact on their teaching style.

These findings align with the conclusions drawn by Keis et al. [34], which emphasize the flexibility of online learning compared to traditional face-to-face mode, which tends to be more time and resource intensive. Aligning with Akyildiz [35] and Domenici [36], online education has been essential for academic continuity during institutional closures. This support should extend beyond technical assistance and encompass pedagogical aid for both educators and students. As a result of receiving sufficient technical support, students and educators expressed their appreciation for the support provided by their institutions. This is particularly crucial since technical support is essential for educators to effectively utilize digital tools. Koh and Daniel [4] reported that it crucial to ensure the sustainability of pedagogical innovation through online learning, not only during the pandemic but also in the future.

### 6.2. Learning Sphere and Examination Methods

Innovation plays a significant role in fostering education, acting as a catalyst for socio-economic development [37]. Digital innovation provides the essential infrastructure for the implementation of online education, thus enabling the emergence of novel teaching and learning paradigms. In this digital era, where technology has become an integral part of education, the frameworks supported by innovation open up a world of possibilities. This transformation in educational models not only affects how knowledge is disseminated but also influences how learners engage with information and interact with instructors. The dynamic nature of digital innovation continuously shapes the educational landscape, fostering adaptability and creative approaches in the teaching–learning process. Furthermore, digital innovation serves as a crucial medium, allowing for the rapid dissemination of information and knowledge to the community, particularly in times of crises like the pandemic. It becomes an indispensable tool for maintaining educational continuity and providing access to vital resources when traditional methods may be disrupted. The synergy of digital innovation and education not only enhances the learning experience but also contributes to the resilience of educational systems in the face of unforeseen challenges [38].

The findings exhibit that students generally expressed a moderate satisfaction with the learning materials and examination methods. This level of contentment implies that students found the diversity in their learning resources to be reasonably sufficient, aligning with the observations of prior research [39]. Digital communication platforms, as discussed in the literature, have proven to be vital assets for educational institutions. These platforms, which have come to the forefront during the COVID-19 pandemic, played an essential role in not only ensuring the survival of educational processes but also in facilitating their growth and enhancement [40]. In parallel, the educators also expressed a moderate level of agreement concerning learning materials and examination methods. This moderation indicates a dedication among educators to effectively impart knowledge, even when confronted with challenging circumstances. Their adaptability and resilience, as evidenced by these findings, are certainly commendable. The educators' ability to maintain their commitment to effective teaching in the face of unprecedented challenges is a positive and notable development.

On the other hand, the lowest ranked item according to students was the reversed item about focus during the online classes, from which we conclude that only a small number of participants do not lose focus during online lessons. Notably also, the item with the lowest rank pertains to examination methods according to the educators. This implies

that educators have legitimate concerns about the fairness and effectiveness of online examinations. One of the difficulties encountered in online education revolved around the creation of student assessments [1]. Such concerns are valid, as online examinations can be challenging to supervise and may not accurately reflect students' true learning outcomes.

### 6.3. Students and Educators' Resilience

In this study, as anticipated, both students and educators demonstrated a high level of agreement regarding their resilience levels. Students were successful in adapting to the new learning environment and maintaining effective learning. Similarly, educators showcased their ability to acquire new skills and adjust their teaching approaches to cater to their students' needs. Enhancing educational resilience can bring about positive outcomes across various domains, including individuals, academic institutions, industries, and society, addressing a wide range of challenges and offering numerous substantial advantages [41].

However, it is worth noting that the lowest-ranked item among students pertained to their satisfaction with teamwork engagement while studying online. This suggests that participants might have encountered difficulties collaborating with their peers in the online setting. Conversely, among educators, the lowest-ranked item related to their ability to tolerate high levels of ambiguity and uncertainty in various situations. This indicates that educators may have faced challenges in dealing with the uncertainties brought about by the pandemic. Such difficulties are understandable, considering that the pandemic posed a novel and demanding experience for everyone. Nonetheless, it is imperative to acknowledge that ambiguity and uncertainty are inherent aspects of life, necessitating that educators cultivate the required capabilities to navigate these challenges adeptly for achieving success in their teaching roles.

As emphasized by Wang et al. [42], significant resources, encompassing resilience and a supportive learning environment, play a pivotal in aiding students. In this context, such aforementioned challenges extend beyond the pandemic and are expected to endure in the ever-evolving educational landscape. This underscores the necessity for ongoing adaptation and innovation in teaching and learning practices, as educators and institutions strive to provide effective support to students in the face of uncertainty.

### 6.4. Online Class Perception

Students demonstrated a notably high level of agreement regarding their perception of online classes, while educators showed a moderate level of agreement in their perception and assessment of online classes. Educators acknowledge the potential benefits of online classes. However, the fact that the lowest-ranked item was "I believe the students learn better through online classes" suggests that educators have reservations about the effectiveness of online learning. This discrepancy suggests that educators may harbor some reservations about the efficacy of online learning, which warrants closer consideration. Such concerns are significant in the context of evolving educational paradigms, particularly as digital learning continues to gain prominence. This dynamic underline the need for further exploration and analysis of the factors influencing educators' perceptions and attitudes toward online education.

According to Oliveira et al., a significant drawback of the emergency remote education system is the absence of direct interaction and communication between the educator and the learner, which is essential for effective learning [28]. This aligns with Lassoued et al. (2020), who identified key challenges to maintaining quality in emergency remote education during the COVID-19 pandemic, which include personal barriers such as student resistance and rejection; pedagogical hurdles related to conducting assessments and obtaining feedback to evaluate student strengths and weaknesses; and technical issues like poor internet connectivity, data security, and website piracy [43].

In the midst of the COVID-19 pandemic, the perceptions of students and educators in higher education regarding online classes underwent a significant shift, as evidenced by the disparity in agreement levels between the two groups. While students displayed a notably

high level of favorability towards online learning, educators exhibited a more moderate perspective, hinting at underlying reservations about its efficacy. This discrepancy in viewpoints underscores a critical area that warrants thorough investigation, unraveling the factors influencing educators' attitudes and apprehensions, particularly concerning the effectiveness of online education. Given the growing prominence of digital learning in educational paradigms, understanding these influential factors becomes pivotal for enhancing the overall efficacy of online education. Hence, this focused examination on the perceptions of educators aims to uncover nuanced insights that can inform and shape the evolution of educational practices, fostering a more robust and adaptable system aligned with global teaching and learning trends.

Given the nuanced nature of perceptions surrounding online classes during the COVID-19 pandemic, we focused our investigation on delving deeper into the online class perception factor. Utilizing chi-squared analysis, we conducted a comprehensive examination and rigorous analysis to discern the underlying dynamics influencing educators' and students' attitudes towards online education. This approach enabled us to extract intricate insights, shedding light on the multifaceted nature of educators' and students' perceptions and reservations regarding the effectiveness of online learning.

The Pearson chi-squared test was used for further analysis. This statistical method is utilized to assess whether a noteworthy connection exists between two categorical variables. The null hypothesis (H0) posits that there is no association between these variables, while the alternative hypothesis (H1) suggests that a meaningful connection indeed exists. The test statistic is computed by contrasting the actual frequencies within each category of the contingency table with the anticipated frequencies based on the null hypothesis. The *p*-value signifies the probability of obtaining a test statistic as extreme as the observed one or even more extreme, assuming the null hypothesis to be true. If the *p*-value falls below the significance threshold (typically 0.05), we reject the null hypothesis and affirm the presence of a significant association between the two variables.

**H0.** *There is no significant association between the two variables at level 0.05.*

**H1.** *There is a significant association between the two variables at level 0.05.*

### 6.5. Educators' Perception

**H1.** *The educator's experience positively affects the educator's opinions about the hybrid educational process after the pandemic.*

As shown in Table 15, the null hypothesis (H0) is that there is no significant association between the two variables, while the alternative hypothesis (H1) is that there is a significant association between the two variables. In the above table, the *p*-value was 0.748, which is greater than the significance level of 0.05. This means that we cannot reject the null hypothesis and conclude that there was no significant association between educators' professional teaching experience and their preference for the learning process after the pandemic. However, there were some interesting trends in the data. Educators with less than 15 years of experience were found to be slightly more likely to prefer a hybrid learning model, while educators with more than 15 years of experience were more likely to prefer a face-to-face learning. Overall, the data did not provide strong evidence to support the claim that educators' experience has a positive effect on their opinion about hybrid educational process after the pandemic. However, there are were interesting trends in the data that suggest that educators with less experience may be more open to hybrid learning models, as shown in Figure 1.

**Table 15.** Results with the two factors extracted (Educators professional teaching experience).

| How Do You Prefer the Learning Process to be after the Pandemic? | | | Face-to-Face | Hybrid | Online | Total |
|---|---|---|---|---|---|---|
| Professional teaching experience | Less than 5 years | n | 7 | 9 | 0 | 16 |
| | | % | 43.8% | 56.3% | 0.0% | 100.0% |
| | 5 to less than 10 years | n | 7 | 12 | 0 | 19 |
| | | % | 36.8% | 63.2% | 0.0% | 100.0% |
| | 10 to less than 15 years | n | 10 | 11 | 1 | 22 |
| | | % | 45.5% | 50.0% | 4.5% | 100.0% |
| | 15 to less than 20 years | n | 11 | 7 | 0 | 18 |
| | | % | 61.1% | 38.9% | 0.0% | 100.0% |
| | 20 years+ | n | 15 | 12 | 1 | 28 |
| | | % | 53.6% | 42.9% | 3.6% | 100.0% |
| Total | | n | 50 | 51 | 2 | 103 |
| | | % | 48.5% | 49.5% | 1.9% | 100.0% |
| Pearson chi-squared = 5.090, df = 8, *p*-value = 0.748 | | | | | | |

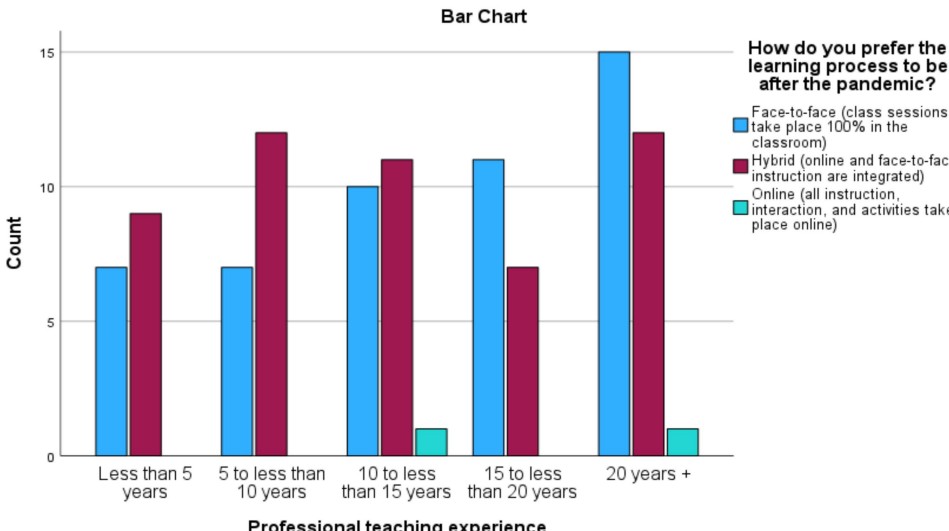

**Figure 1.** Teaching experience's relationship with learning process type: educators' perspective.

**H2.** *The type of college affected educators' opinions on the hybrid educational process after the pandemic.*

The data in Table 16 show no statistically significant relationship between the type of college an educator taught in and their preference for the learning process after the pandemic. The *p*-value was 0.747, which is greater than the significance level of 0.05. This means we cannot reject the null hypothesis that there was no association between the two variables. However, there were some interesting trends in the data. Educators who taught in the College of Health and Life Sciences and the College of Arts and Science were more likely to prefer face-to-face learning by a percentage of 66.7%. In contrast, educators who taught in the College of Education and College of Medicine and Pharmacy were more likely to prefer hybrid learning. Educators who taught in the College of Business and Economics, the College of Law, and the College of Computer Science were evenly split between face-to-face and hybrid by equal percentages of 50% each. Overall, the data did not

provide strong evidence to support the claim that the type of college an educator teaches in affects their opinion about the hybrid educational process after the pandemic. However, some interesting trends in the data suggest that educators in certain colleges may be more likely to prefer certain learning models, as shown in Figure 2.

**Table 16.** Results with the two factors extracted (Educators' teaching field).

| How Do You Prefer the Learning Process to be after the Pandemic? | | | Face-to-Face | Hybrid | Online | Total |
|---|---|---|---|---|---|---|
| What is the college you teach in? | College of Engineering | n | 5 | 7 | 0 | 12 |
| | | %. | 41.7% | 58.3% | 0.0% | 100.0% |
| | College of Arts and Science | n | 8 | 4 | 0 | 12 |
| | | %. | 66.7% | 33.3% | 0.0% | 100.0% |
| | College of Humanities and Social Sciences | n | 7 | 6 | 1 | 14 |
| | | %. | 50.0% | 42.9% | 7.1% | 100.0% |
| | College of Business and Economics | n | 7 | 7 | 0 | 14 |
| | | %. | 50.0% | 50.0% | 0.0% | 100.0% |
| | College of Education | n | 3 | 8 | 1 | 12 |
| | | %. | 25.0% | 66.7% | 8.3% | 100.0% |
| | College of Health and Life Sciences | n | 4 | 2 | 0 | 6 |
| | | %. | 66.7% | 33.3% | 0.0% | 100.0% |
| | College of Medicine and Pharmacy | n | 2 | 6 | 0 | 8 |
| | | %. | 25.0% | 75.0% | 0.0% | 100.0% |
| | College of Law | n | 5 | 5 | 0 | 10 |
| | | %. | 50.0% | 50.0% | 0.0% | 100.0% |
| | College of Computer Science | n | 3 | 3 | 0 | 6 |
| | | %. | 50.0% | 50.0% | 0.0% | 100.0% |
| | Other | n | 6 | 3 | 0 | 9 |
| | | %. | 66.7% | 33.3% | 0.0% | 100.0% |
| | Total | n | 50 | 51 | 2 | 103 |
| | | %. | 48.5% | 49.5% | 1.9% | 100.0% |
| Pearson chi-squared = 13.722, df = 18, *p*-value = 0.747 | | | | | | |

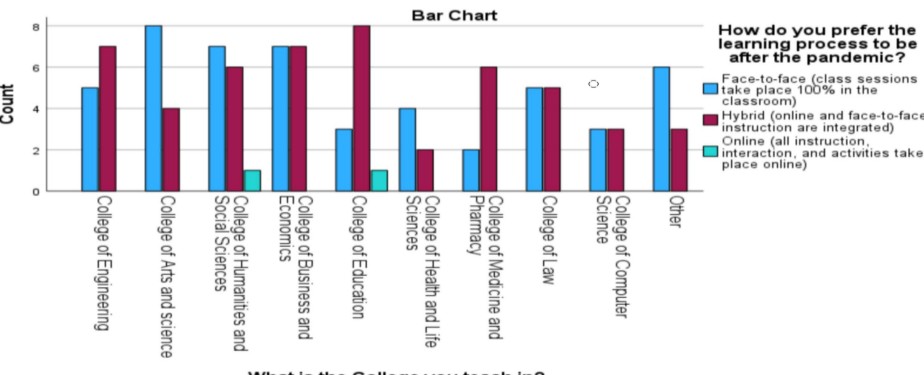

**Figure 2.** Collage with relation to learning process type: educators' perspective.

*6.6. Students' Perception*

**H1.** *The type of college affected students' opinions about the hybrid educational process after the pandemic.*

The data in Table 17 show that there was a statistically significant relationship between the type of college a student attended and their preference for the learning process after the pandemic. The *p*-value was 0.026, which is less than the significance level of 0.05. This means that we reject the null hypothesis that there was no association between the two variables and accept the alternative hypothesis that there was significant association between the two variables. Specifically, students in the College of Engineering, College of Arts and Science, College of Humanities and Social Sciences, College of Business and Economics, College of Law, College of Medicine and Pharmacy, and College of Computer Science were more likely to prefer hybrid learning. College of Education and Health and Life Sciences students were more likely to choose face-to-face learning. Students in the College of Sharia and Islamic Studies and other colleges were evenly split between face-to-face, hybrid, and online learning. Overall, the data provide strong evidence to support the claim that the type of college a student attends affects their opinion about the hybrid educational process after the pandemic. As shown in the Figure 3, students in certain colleges are more likely to prefer certain learning models.

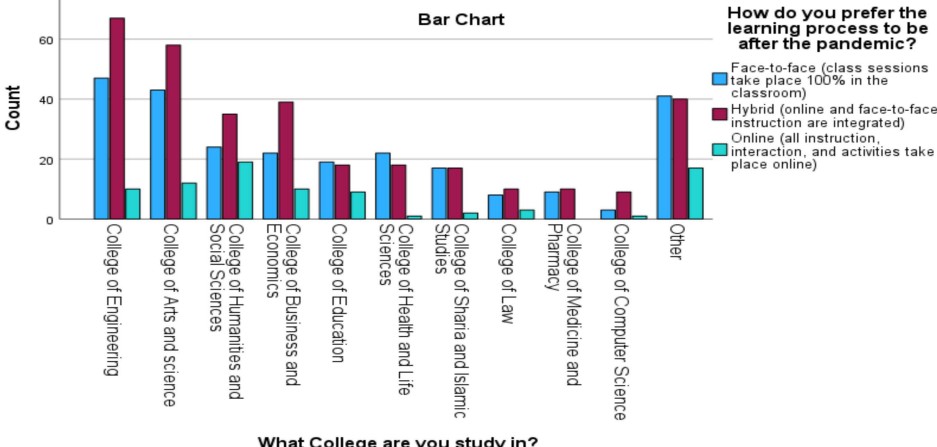

**Figure 3.** College relationship with learning process type: students' perspective.

**H2.** *Full-time students have a positive opinion about the hybrid educational process after the pandemic.*

The data in Table 18 show that there was a statistically significant relationship between whether a student was a full-time employee and their preference for the learning process after the pandemic. The *p*-value was 0.000, which is less than the significance level of 0.05. This means that we can reject the null hypothesis that there was no association between the two variables. Specifically, students who were not full-time employees were more likely to prefer face-to-face learning at 41.4%, while students who were full-time employees were more likely to prefer hybrid online learning at 47.4%. This is likely because full-time employees have less time to attend in-person classes, so they may prefer a more flexible learning model. Overall, the data provide strong evidence to support the claim that full-time employment status affects students' opinions about the hybrid educational process after the pandemic. As shown in the Figure 4, full-time employees are more likely to prefer hybrid or online learning models.

**Table 17.** Results with the two factors extracted (Students' studying field).

| How Do You Prefer the Learning Process to be after the Pandemic? | | | Face-to-Face | Hybrid | Online | Total |
|---|---|---|---|---|---|---|
| | College of Engineering | n | 47 | 67 | 10 | 124 |
| | | % | 37.9% | 54.0% | 8.1% | 100.0% |
| | College of Arts and Science | n | 43 | 58 | 12 | 113 |
| | | % | 38.1% | 51.3% | 10.6% | 100.0% |
| | College of Humanities and Social Sciences | n | 24 | 35 | 19 | 78 |
| | | % | 30.8% | 44.9% | 24.4% | 100.0% |
| | College of Business and Economics | n | 22 | 39 | 10 | 71 |
| | | % | 31.0% | 54.9% | 14.1% | 100.0% |
| | College of Education | n | 19 | 18 | 9 | 46 |
| | | % | 41.3% | 39.1% | 19.6% | 100.0% |
| What college are you studying in? | College of Health and Life Sciences | n | 22 | 18 | 1 | 41 |
| | | % | 53.7% | 43.9% | 2.4% | 100.0% |
| | College of Sharia and Islamic Studies | n | 17 | 17 | 2 | 36 |
| | | % | 47.2% | 47.2% | 5.6% | 100.0% |
| | College of Law | n | 8 | 10 | 3 | 21 |
| | | % | 38.1% | 47.6% | 14.3% | 100.0% |
| | College of Medicine and Pharmacy | n | 9 | 10 | 0 | 19 |
| | | % | 47.4% | 52.6% | 0.0% | 100.0% |
| | College of Computer Science | n | 3 | 9 | 1 | 13 |
| | | % | 23.1% | 69.2% | 7.7% | 100.0% |
| | Other | n | 41 | 40 | 17 | 98 |
| | | % | 41.8% | 40.8% | 17.3% | 100.0% |
| | Total | n | 255 | 321 | 84 | 660 |
| | | % | 38.6% | 48.6% | 12.7% | 100.0% |
| Pearson chi-squared = 33.999, df = 20, *p*-value = 0.026 | | | | | | |

**Table 18.** Results with the two factors extracted (Students employment status).

| How Do You Prefer the Learning Process to be after the Pandemic? | | | Face-to-Face | Hybrid | Online | Total |
|---|---|---|---|---|---|---|
| Are you currently a full-time employee? | No | n | 179 | 213 | 40 | 432 |
| | | % | 41.4% | 49.3% | 9.3% | 100.0% |
| | Yes | n | 76 | 108 | 44 | 228 |
| | | % | 33.3% | 47.4% | 19.3% | 100.0% |
| | Total | n | 255 | 321 | 84 | 660 |
| | | % | 38.6% | 48.6% | 12.7% | 100.0% |
| Pearson chi-squared = 14.468, df = 2, *p*-value = 0.000 | | | | | | |

In the midst of the COVID-19 pandemic, the perceptions of students and educators in HE regarding online classes underwent a significant shift. Notably, students displayed a high level of agreement in their favorable perception of online learning, while a more moderate level of understanding marked educators' attitudes. This disparity in viewpoints suggests that educators may harbor reservations about the effectiveness of online education, a topic that merits closer scrutiny. This contrast underscores the importance of delving

deeper into the factors influencing educators' attitudes and reservations, particularly as digital learning continues to gain prominence in educational paradigms.

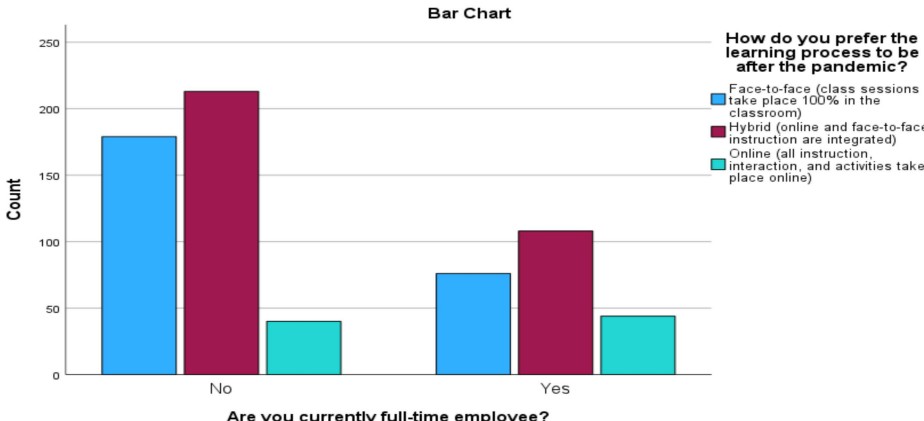

**Figure 4.** Working relation with learning process type: students' perspective.

One crucial aspect highlighted in the review on educational system resilience during the COVID-19 pandemic [14] is the nuanced nature of perceptions surrounding online classes, especially regarding pedagogy and examination methods. The differing viewpoints, especially regarding the effectiveness of online learning, indicate the need for a more comprehensive examination of the factors influencing educators' attitudes. Understanding these factors is paramount for enhancing the overall efficacy of online education, particularly as digital learning continues to gain prominence. Consequently, there is an urgent need for sustained research and dialogue to inform the evolution of educational paradigms.

Figures 5 and 6 summarize the paper's findings about student's and educators' perceptions regarding hybrid education in HE. Educational procedures in HE must align with global teaching and learning trends. The pandemic accelerated the adoption of online education, representing a complex and evolving method in the educational process. To facilitate this transition, policymakers in education must proactively work on creating policies and related procedures for digitalized education. Such policies should provide clear guidelines for utilization and regulatory measures to ensure accreditation for this educational strategy. The ultimate goal is to create an educational system that transcends physical borders, allowing learning to occur anytime and anywhere, asynchronously and synchronously.

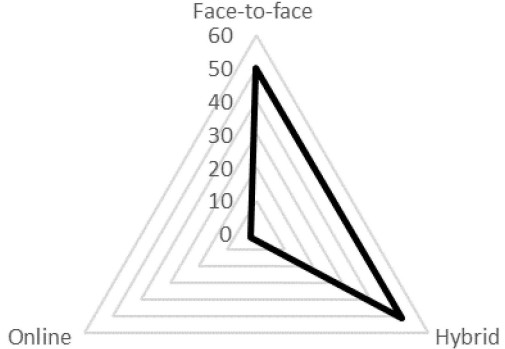

**Figure 5.** Educators' opinions about the learning process.

How do you prefer the learning process to be after the pandemic?

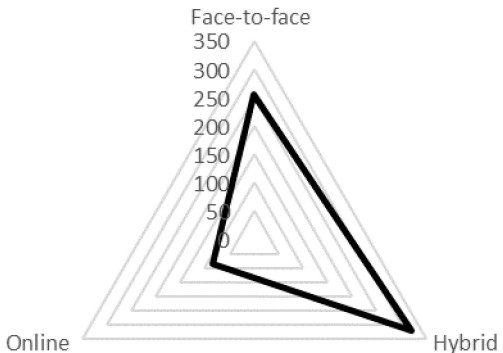

**Figure 6.** Student's opinions about the learning process.

## 7. Conclusions

The impact of the COVID-19 pandemic on higher education has been complex and widespread, with variations observed among institutions and countries. Higher education institutions, as well as their faculties and students, have made substantial efforts to demonstrate resilience and adapt swiftly to the profound changes brought about by the pandemic. Despite heightened protocols and restrictions, the higher education systems in Qatar, as a case study, have played a vital role in actively seeking solutions and alternatives to ensure the uninterrupted progression of the learning process. In this comprehensive exploration of the impact of the COVID-19 pandemic on online education in Qatar, we delved into crucial aspects, shedding light on the experiences of both educators and students. The transformations induced by the pandemic have revolutionized traditional teaching and learning methods, driving the adoption of online platforms across EI, not just in Qatar, but globally. Our analysis covered five key areas: educational platforms, digital tools and support, the learning environment and examination methods, resilience levels during the transition to online education, and perceptions of online classes.

In this research, we explored the pivotal role of sustainable education in shaping the future of educational paradigms, mainly through the lens of Education for Sustainable Development (ESD). The study, set against the backdrop of the COVID-19 pandemic's impact on higher education, particularly in Qatar, delves into the swift transition to online education. This shift underscores the critical need for institutional resilience, the strategic incorporation of digital tools, and the adaptability of educators and learners alike. The findings illuminate the dual nature of challenges and opportunities presented by online education, such as refining assessment methodologies and fostering a culture of continuous innovation in pedagogical approaches. These insights are integral to advancing ESD, highlighting the necessity for educational systems to be adaptable, technologically adept, and resilient, thereby contributing to the sustainability of future education.

Students and educators expressed a notable satisfaction with the digital tools and support, recognizing their positive impact. This alignment echoes prior research emphasizing the flexibility of online learning, with digital tools playing a crucial role in ensuring academic continuity. Notably, institutional support, extending beyond technical assistance to pedagogical guidance, emerged as a critical factor in fostering a positive online learning experience. The digitalization of education, while facilitating academic continuity, has presented challenges, including concerns about the fairness and effectiveness of online examinations and assessments. The educators' commendable adaptability and resilience in maintaining effective teaching practices, even amidst challenges, mark a positive trend.

Both students and educators showcased high levels of resilience in adapting to the new learning environment. However, challenges were evident, such as difficulties in teamwork engagement among students and educators grappling with ambiguity and uncertainty. These challenges, inherent in the educational landscape, underscore the need for continuous adaptation and innovation in teaching practices on a global level

as well. Students demonstrated a high level of agreement regarding their perception of online classes, while educators exhibited a more moderate stance. The reservation among educators about the effectiveness of online learning, as indicated by the lowest-ranked item, calls for deeper exploration into the factors shaping educators' attitudes toward online education.

In conclusion, our study illuminates the evolving landscape of education in Qatar amidst the challenges posed by the pandemic. It calls for sustained efforts in leveraging technology for effective teaching, cultivating resilience among educators and students, as well as navigating the uncertainties of the educational landscape with adaptability and innovation. As we forge ahead, the lessons learned from this transformative period will undoubtedly shape the future of education in Qatar and beyond.

## 8. Implications, Limitations, and Future Work

The implications of this study extend beyond the immediate insights gained into the experiences of educators and students in Qatar's higher education system during the COVID-19 pandemic. First and foremost, the findings underscore the critical role of institutional support in facilitating a positive online learning experience. Recognizing the satisfaction expressed by both students and educators with digital tools and support, institutions should consider investing in comprehensive support mechanisms that extend beyond technical assistance to encompass pedagogical guidance. This highlights the importance of a holistic approach in ensuring the successful integration of online education. Moreover, this study sheds light on the challenges faced in the digitalization of education, particularly concerning the fairness and effectiveness of online examinations. The moderate satisfaction expressed by students and educators in this regard calls for a reevaluation of assessment methods in the online learning environment. Educational institutes are prompted to explore innovative approaches to maintain the integrity of assessments while accommodating the dynamic nature of digital education.

The resilience exhibited by both students and educators in adapting to the new learning environment serves as a testament to the potential of flexible and innovative teaching practices. As such, institutions are encouraged to foster a culture of adaptability and continuous learning among their academic community. This adaptability is crucial, not only during times of crisis but as an ongoing strategy to navigate uncertainties in the ever-evolving educational landscape. Furthermore, the study highlights the nuanced perceptions of online classes among students and educators. The disparity in perception, particularly regarding the effectiveness of online learning, signals the need for a deeper exploration of factors influencing educators' attitudes. Understanding these factors is essential for enhancing the efficacy of online education, especially as digital learning continues to gain prominence. This dynamic emphasizes the importance of ongoing research and dialogue to inform the evolution of educational paradigms. This work elaborates to provide insights into the digital transformation of higher education in Qatar during the pandemic, a case hitherto unexplored. The lessons drawn from this research carry implications for future educational practices. Recognizing the centrality of technology in education, institutions must strategically leverage digital tools, fostering a supportive and resilient learning environment.

### *Limitations and Future Directions*

While our study provides valuable insights, it is not without limitations. The unique context of Qatar's higher education system and the specificity of the pandemic period may limit generalizability. Thus, future research should aim to diversify samples to enhance the generalizability of findings. Moreover, the study focuses on the pandemic period, a time marked by unique challenges and adaptations in the education system. The dynamism of the pandemic situation and its impact on teaching and learning may not fully represent the ongoing or future challenges faced by educational institutions. However, these limitations open the door for future research endeavors aimed at broadening our understanding of

resilient education systems. As we move forward, there is a critical need for more in-depth investigations into the intricacies of adapting and thriving in crisis situations. We propose future research to delve into creating a roadmap for a resilient education system, identifying the minimum requirements to characterize such resilience universally. By exploring these aspects, we aim to contribute to the development of robust strategies that can withstand unforeseen challenges and foster educational continuity. This endeavor becomes particularly crucial as we consider the ever-evolving technological landscapes and the lasting impacts the pandemic has left on education globally.

**Supplementary Materials:** The following supporting information can be downloaded at https://www.mdpi.com/article/10.3390/su16062265/s1. The supplementary materials contains 3 parts: supplementary materials S1, S2 and S3, in which there are Tables S1–S12.

**Author Contributions:** Conceptualization, N.M.A., M.Y., and B.C.M.; methodology, N.M.A.; formal analysis, N.M.A., M.Y., and B.C.M.; investigation, N.M.A. and M.Y.; resources, N.M.A. and M.Y.; writing—original draft preparation, N.M.A. and M.Y.; writing—review and editing, N.M.A., M.Y., and B.C.M.; supervision, B.C.M.; project administration, M.Y. and B.C.M. All authors have read and agreed to the published version of the manuscript.

**Funding:** This research received no external funding.

**Institutional Review Board Statement:** This study has obtained the necessary ethical approval from the ethics boards of the participating universities. The research adheres to principles such as voluntary participation, the right to withdraw at any point, and the confidentiality of participants' identities. HBKU serves as the primary host institution for this study, and the survey has received ethical approval from the Hamad Bin Khalifa University—Institutional Review Board (HBKU-IRB) with the IRB Protocol Reference Number (QBRI-IRB-2024-4).

**Informed Consent Statement:** Informed consent was obtained from all subjects involved in the study.

**Data Availability Statement:** This study, approved by the ethics boards of the participating universities, follows ethical principles including voluntary participation, the right to withdraw, and participant identity confidentiality. The primary host institution, HBKU, received ethical approval from the Institutional Review Board (HBKU-IRB) with the protocol reference number QBRI-IRB-2024-4. All survey questions and result tables can be found in the Supplementary Materials file.

**Conflicts of Interest:** The authors declare no conflicts of interest.

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
