# Peer review of "Online Learning and Teaching during the COVID-19 Pandemic in Higher Education in Qatar"

_sustainability, doi:10.3390/su16062265_

Round 1

Reviewer 1 Report

Comments and Suggestions for Authors

This is a timely and relevant study, which will contribute to the growing body of literature on lessons learned about online teaching and learning during and since the pandemic. 

Comments on the Quality of English Language

My biggest concern is that in the Introduction section, the phrasing of some sentences gives the impression that the pandemic state -- and related online learning mandates -- are ongoing. This should be revised to reflect the fact that many (if not most) higher education institutions have returned to relatively normal operations. 

Author Response

This is a timely and relevant study, which will contribute to the growing body of literature on lessons learned about online teaching and learning during and since the pandemic. 

  • My biggest concern is that in the Introduction section, the phrasing of some sentences gives the impression that the pandemic state -- and related online learning mandates -- are ongoing. This should be revised to reflect the fact that many (if not most) higher education institutions have returned to relatively normal operations. 

Thank you for your valuable feedback. We've revised the language in the whole paper to ensure clarity. Additionally, we've included more explanatory background to enhance contextual understanding. Your input has significantly contributed to many improvements and has been instrumental in streamlining the content to better serve the study's objectives. We genuinely appreciate your thoughtful recommendations.

Reviewer 2 Report

Comments and Suggestions for Authors

First, I would like to thank you for the opportunity to read this work. I found this study interesting and with relevant practical implications for the experiences of both students and educators concerning the use of digital tools.

I just have some suggestions:

-          It is unnecessary to explain in detail what reliability, Cronbach alpha, and validity consist of. The same concerns SPSS.

-          Also, the method and results section are difficult to follow the reasoning behind. The authors need to clarify each step conducted better and narrow the paper's focus, considering the study's main goals.

-          The authors highlight the importance of analyzing the specific case of Qatar. It will be important to add a comparison with some other countries to strengthen the uniqueness of the study conducted. Is that context so different from what happened in European countries, for instance?

Comments on the Quality of English Language

Minor editing of English language required.

Author Response

First, I would like to thank you for the opportunity to read this work. I found this study interesting and with relevant practical implications for the experiences of both students and educators concerning the use of digital tools.

I just have some suggestions:

  • It is unnecessary to explain in detail what reliability, Cronbach alpha, and validity consist of. The same concerns SPSS.

Thank you for your valuable insights into our study. We've taken your suggestions into account and removed some lines  (224-234), (236-238), (244-249), (306-310), and    (316-324) that detailed the concepts of reliability, Cronbach alpha, validity, and SPSS. This revision aims to enhance the study's precision, aligning with your advice on focusing more directly on the practical implications for students and educators regarding online education.

  • Also, the method and results section are difficult to follow the reasoning behind. The authors need to clarify each step conducted better and narrow the paper's focus, considering the study's main goals.

Thank you for the valuable comment.

The reasoning behind the method followed in the paper is to investigate and comprehend the impact of the COVID-19 pandemic on the education system, specifically within the context of Qatar. The study aims to explore the experiences and perceptions of educators and students in adapting to online education during the crisis. This exploration is designed to shed light on various key areas such as the educational platform, digital tools, learning environments, examination methods, resilience levels during the transition, and perceptions of online classes. The method used involves conducting an online survey targeting educators and students in three selected universities in Qatar to capture their experiences and perspectives. This methodological approach is guided by the objective of obtaining comprehensive insights into the challenges, support mechanisms, and adaptation strategies employed during the transition to online education, aligning closely with the study's specified research goals and objectives.

The following sentences have been added in lines (186-189) to add more clarifications and highlight the direct link between the study's objectives, the methodology, and results. “To ensure clarity in the methodology's alignment with the study's goals, the subsequent section delineates a comprehensive overview of the empirical investigation approach and the specific areas of focus”. “The methodology employed in this study has been designed to precisely capture the experiences and perceptions of educators and students, ensuring a direct correlation with the outlined research objectives, as detailed in the Introduction”.

  • The authors highlight the importance of analyzing the specific case of Qatar. It will be important to add a comparison with some other countries to strengthen the uniqueness of the study conducted. Is that context so different from what happened in European countries, for instance?

Thank you for your insightful comments. We acknowledge the significance of contextualizing our study within a broader international perspective. The specific case of Qatar in our research provides a detailed examination of the challenges and adaptations in the education system during the COVID-19 pandemic. Our primary focus lies in comprehensively exploring the experiences and responses within the Qatari educational context, considering its unique socio-cultural and institutional framework. While we appreciate the suggestion to include a comparison with other countries, the scope of our study is primarily centered on Qatar's educational landscape. The complexities of educational transitions during the pandemic vary significantly across regions, and while a comparative analysis with European countries could indeed be valuable, it falls outside the intended scope and depth of our investigation. Nonetheless, we believe that future research initiatives could delve into comparative analyses across various global contexts to offer a more extensive understanding of international educational responses to crises like the pandemic. Your input has prompted thoughtful considerations for future studies exploring broader cross-country comparisons in the field of pandemic-induced educational adaptations.

However, we added some motivational sentences in the introduction in lines (71-82) [Qatar's national education landscape serves as a significant foundation in online education in higher education. for the research we are presenting. It is crucial to note that Qatar places significant importance on education, viewing it as a fundamental component of the country's development and a top national priority, as recognized by the General Secretariat for Development Planning in 2008.In the context of Higher Education Institutions (HEIs), it becomes apparent that universities have excelled in providing robust support systems during emergency remote teaching. The university has demonstrated remarkable proficiency in delivering technical assistance. Drawing parallels with similar situations in universities across Australia, New Zealand, and Turkey, three distinct case scenarios reveal striking commonalities. These scenarios emphasize the vital role of leadership initiatives essential for higher education transformation in times of a pandemic crisis.].

Your inputs have been instrumental in streamlining the content to better serve the study's objectives. We genuinely appreciate your thoughtful recommendations.

Reviewer 3 Report

Comments and Suggestions for Authors

This study addresses the issue of e-learning. The authors focused on five 5 primary domains of this paper: the educational platform, digital tools and the corresponding support mechanisms, the learning environment and assessment methodologies, the level of adaptability exhibited during the shift to online education, the perceptions of online classes among both students and educators in the context of the pandemic-driven transition to online education. While, there are some ambiguous points needed to clarify in this paper.

1.      There are various he related studies in this field, while the literature review is so limited in your Introduction section. We expect this study can find something unique and is associated the knowledge of the field.

2.      The sampling technique is unclear in this study in terms of population and fitted sample size needed to define.

3.      Table 1, why gender gap is so big? Did it fit to the structure of target population?

4.      In Table 2, the diploma of education is unclear, please clarify.

5.      Since the students and faculty group are important, why the comparison is neglect?

6.      The Remarks section should display ahead of Discussion section. Moreover, the hypotheses may display after reviewing related previous studies or theories.

7.      Citations of this study is not enough, at this moment, there are a lot of studies in this field. This part should be enhanced.

Author Response

All in the file

Reviewer 4 Report

Comments and Suggestions for Authors

My opinion is major revision, and my suggestions are as follows:

The research method is relatively simple and lacks detailed analysis and explanation.

The background factors of students are not taken into account, which may lead to conclusions that are not accurate enough.

The study focused only on the experiences of students and teachers in online education, but did not compare with other forms of education, such as traditional classroom education.

Comments on the Quality of English Language

nothing

Author Response

All in the file

Round 2

Reviewer 2 Report

Comments and Suggestions for Authors

The authors addressed my previous issues properly. As such, I have nothing more substantial to add. 

Author Response

The authors have no questions to address. 

Reviewer 3 Report

Comments and Suggestions for Authors

1. The authors have made significant revisions to this version. Most of my questions have been answered.

2. This survey explored undergraduate and graduate students and educators at Hamad Bin Khalifa Uni- 189 varsity (HBKU), Qatar University (QU), and the Community College of Qatar (CCQ). The addressing of the case study and its process was well organized.

3. The results follow five domains: platforms, digital tools and support, the learning environment and examination methods, resilience levels during the transition to online education, and perceptions of online classes. In comparison, the discussion section did not. For example, the faculty’s view on “I believe the students learn better through online classes” is low. It is an important finding. Was it similar to previous studies? Or it is only a unique phenomenon in the case study.

4. Table 5, students’ opinion:

What type of attendance do you prefer after the pandemic? The findings reveal that virtual (online class) is a small part, but most students prefer Physical (in-person) and a Combination of both. It is interesting.

5. Table 6, educators’ opinion:

What do you prefer the learning process to be after the pandemic?

Most educators tend to Face-to-face (class sessions take place 100% in the classroom) and Hybrid (online and face-to-face instructions are integrated). This phenomenon may need to be clarified and interpreted.
